# BENCH-CoE: A FRAMEWORK FOR COLLABORATION OF EXPERTS FROM BENCHMARK

## ABSTRACT

Large Language Models (LLMs) are key technologies that drive intelligent systems to handle multiple tasks. To meet the demands of various tasks, an increasing number of LLMs-driven experts with diverse capabilities have been developed, spreading from language to visual understanding and generalization, accompanied by corresponding benchmarks to evaluate their performance. This paper proposes the Bench-CoE framework, which enables Collaboration of Experts (CoE) by effectively leveraging benchmark evaluations to achieve optimal performance across various tasks. Bench-CoE consists of a set of specialized expert models, a router for assigning tasks to corresponding experts, and a benchmark dataset for training the router. Based on this framework, we first formulate Query-Level Bench-CoE that is an abstraction of existing CoE methods exploiting the benchmark dataset. We further propose Subject-Level Bench-CoE, a new method that effectively addresses the potential issues of Query-Level Bench-CoE in poor generalization and labeling costs during training the router. Experiments show that the Query-Level Bench-CoE excels in in-distribution tasks, while the Subject-Level Bench-CoE demonstrates stronger out-of-distribution generalization and cross-domain scenarios adaptability. The codes are available at: https://anonymous.4open.science/r/BenchCoE.

## 1 INTRODUCTION

Large Language Models (LLMs) are capable of performing various natural language processing (NLP) tasks through auto-regressive prediction conditioned on task prompts Radford et al. (2019); Brown et al. (2020). The ability of LLMs to describe and unify tasks makes them key components in current visual understanding tasks, leading to the emergence of Large Multimodal Models (LMMs) Liu et al. (2023); Zhu et al. (2024). While these LLMs/LMMs-driven experts can handle a wide range of visual and language tasks, they possess different areas of expertise and exhibit significant performance variations across different tasks. As the capabilities of experts continue to improve, benchmark evaluations have also become increasingly complex and diverse Tjong Kim Sang (2002); Bowman et al. (2015); Rajpurkar et al. (2016). For instance, benchmark such as MMLU Wang et al. (2024) is used to assess multi-subject reasoning abilities in language tasks, while MMMU Yue et al. (2024) evaluates cross-domain reasoning in multimodal tasks. It is difficult for a single model to achieve optimal performance across all tasks. Moreover, attaining comparable effectiveness often requires an increase in model scale and inference cost, which in turn poses challenges for computational resources and practical applications.

To integrate the advantages of the model in various aspects while controlling costs, researchers have proposed different approaches. Mixture of Experts (MoE) Jacobs et al. (1991); Shazeer et al. (2017); Fedus et al. (2022) introduces multiple expert models and adopts a sparse activation strategy, where only a subset of experts is activated during inference, thereby reducing computational cost. However, the experts in MoE is just sub-modules of a model, and it is difficult in understanding the functionality of each expert and the decision-making process in solving certain tasks. To mitigate this issue, researchers proposed Collaboration of Experts (CoE) Zhang et al. (2022); Jiang et al. (2023b); Ong et al. (2024); Lu et al. (2024), which integrates the advantages of different models by routing query to specific expert model, typically implemented through an expert router. However, these CoE methods rely on additional labeled data to train the router, and their generalization ability is often limited when handling out-of-distribution (OOD) tasks, posing challenges for expanding the

capabilities of CoE models. Is there a way to maintain generalization without the additional cost of labeled data? The evaluation results of various experts across different subjects in the benchmark shed light on the answer to the problem.

We analyzed that the key to the problem lies in obtaining reasonable labels for query assignment, and the performance of expert models across different subjects in benchmark tests can actually serve as a type of label. Inspired by this, we propose the Bench-CoE framework, which enables experts collaboration by effectively leveraging the strengths of different experts from benchmark evaluations, as shown in Figure 1. Bench-CoE consists of a set of specialized expert models, a router for expert assignment, and a benchmark dataset for training the router. Based on this framework, we first formulate the Query-Level Bench-CoE which is an abstraction of previous methods, such as Shnitzer et al. (2023); Stripelis et al. (2024). However, this approach requires re-evaluating the performance of different expert models for each query to create data labels. Although the router can effectively allocate in-distribution queries to the appropriate experts, it is expensive in terms of labels creation and struggles to generalize to tasks outside the data distribution.

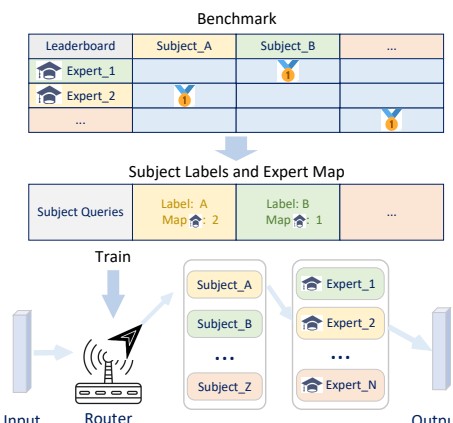

Figure 1: The framework of Bench-CoE. It consists of a set of specialized expert models, a router for expert assignment, and a benchmark dataset for training the router.

To address this issue, we further propose Subject-Level Bench-CoE, which effectively exploits Subject-Level label from benchmark evaluations. The router of Subject-Level Bench-CoE includes a subject classifier and a subject-expert mapping. Firstly, the category index of the subject is used as the category label for all queries within that subject and is utilized to train the query subject classifier. Then, the subject-expert mapping is established by identifying which expert excels in a specific subject from benchmark. Defining the above subject-level labels and subject-expert mapping is effortless because some existing benchmarks Wang et al. (2024); Yue et al. (2024) typically provide subject labels along with evaluation results for each subject. During testing, the input query is first classified into a subject using the subject classifier. Then, based on the expert-subject mapping, the query is assigned to the expert which is most proficient in that subject for processing.

We conducted a series of experiments to evaluate the effectiveness and generalization ability of the Bench-CoE framework. The experimental results show that the query-level router performs better on in-distribution tasks, while the subject-level router demonstrates stronger generalization ability on out-of-distribution data, showcasing better adaptability and robustness. In summary, our main contributions are as follows:

- We propose a simple and efficient framework Bench-CoE for combining and assigning LLM/LMM-driven experts, which achieves flexible and efficient routing without relying on extensive labeled data and large-scale training.
- We propose Subject-Level Bench-CoE which utilizes subject-level labels from benchmark to train a subject classifier, enabling queries to be assigned to the most proficient expert based on an expert-subject mapping.
- Experiments demonstrate that our proposed CoE method outperforms individual models in multi-task scenarios, enhancing cross-domain multi-task processing performance with stronger generalization.

## 2 RELATED WORK

With the rapid proliferation of expert models across various domains, an increasing number of studies focus on efficiently leveraging their expertise. To this end, diverse architectural paradigms have been proposed to balance performance and efficiency. Among them, Mixture of Experts (MoE) and Collaboration of Experts (CoE) are two predominant paradigms, implementing expert routing at the token level and query level, respectively.

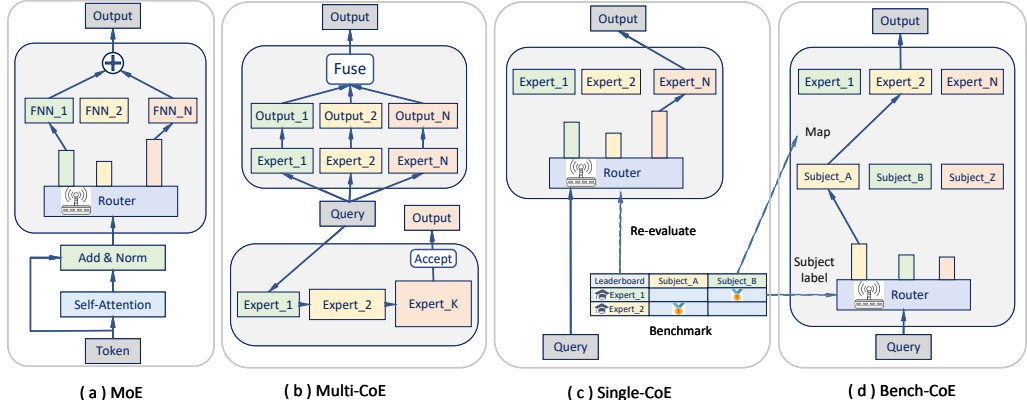

Figure 2: (a) MoE employs multiple experts during inference and aggregates their outputs to predict the output. (b) Multi-CoE includes Parallel-CoE and Cascade-CoE, which respectively utilize multiple parallel experts to generate and integrate their individual outputs, or process sequentially through serial experts until an acceptable output is obtained. (c) The Single-CoE predicts each expert's score on a given query and selects the expert with the highest score to process the query. (d) Bench-CoE includes Query-Level and Subject-Level Bench-CoE, with the latter using benchmark subject-level labels and expert-subject mapping to assign query to the most proficient expert.

## 2.1 MIXTURE OF EXPERTS

MoE primarily employs a sparse activation mechanism, dynamically selecting sub-models (experts) for input query, thereby enabling large-scale parameter expansion while maintaining stable computational costs, as shown in Figure 2 (a). The concept of MoE was first introduced by Jacobs et al., initially focusing on small-scale models and relying on a gating network to select the most suitable expert Jacobs et al. (1991). As computational resources have increased, Shazeer et al. proposed Sparsely-Gated MoE, which was the first to adopt the MoE structure in large-scale deep learning tasks and demonstrated significant performance improvements in neural machine translation Shazeer et al. (2017). However, this method still faced challenges such as imbalanced expert assignment. To address these issues, Lepikhin et al. introduced GShard, which alleviated token load balancing through auxiliary loss, random routing and expert capacity constraints Lepikhin et al. (2020) . Subsequently, Fedus et al. proposed Switch Transformer, which further optimized the MoE routing algorithm and successfully trained trillion-parameter MoE language models Fedus et al. (2022). More recently, Zoph et al. introduced V-MoE, incorporating MoE mechanisms into computer vision (CV) tasks and achieving state-of-the-art results on the image classification task Riquelme et al. (2021). However, each individual expert module is part of MoE and cannot operate independently out of MoE, making it difficult to interpret the role of each expert.

## 2.2 COLLABORATION OF EXPERTS

To address these limitations, researchers have explored an alternative approach CoE which dynamically assigns query to the most suitable expert model. Current methods can be categorized based on their inference approach into two types: Multi-Inference CoE and Single-Inference CoE.

**Multi-Inference CoE** requires multiple models to process the input query and merge all output results, as shown in Figure 2 (b). LLM-BLENDER Jiang et al. (2023b) utilizes multiple different expert models to generate multiple candidate outputs, then employs the pair-ranker module to perform pairwise comparisons and select the top-ranked candidates, followed by the gen-fuser module to generate a high-quality final output. Distinct from this approach, FrugalGPT Chen et al. (2023) explores three strategies for reducing LLM inference costs: Prompt Adaptation, LLM Approximation, and LLM Cascade. It sequentially utilizes progressively more powerful expert models to process the input task until a satisfactory response is obtained. Although this approach can achieve satisfactory results, it incurs a high computational cost as it requires multiple expert inference.

**Single-Inference CoE** route query to a single most suitable expert model for processing, as shown in Figure 2 (c). This approach requires additional datasets to generate query-level routing training labels. The CoE method Zhang et al. (2022) employs a delegator mechanism combined with WGM and LGM training algorithms to improve the adaptability of expert models. Additionally, CCoE

Huang et al. (2024) utilizes a shared backbone with expert sub-networks for efficient expert integration and introduces rule-based gating mechanisms and expert planning. GraphRouter Feng et al. (2024) and Eagle Zhao et al. (2024) leverage heterogeneous graph learning and global and local ELO ranking mechanisms, respectively, to optimize expert model selection strategies. ZOOTER Lu et al. (2024) and TO-Router Stripelis et al. (2024) improve expert selection by employing reward distillation for training routing functions and an intelligent routing mechanism to precisely match queries with expert LLMs. RouteLLM Ong et al. (2024) and Hybrid LLM Ding et al. (2024) reduce inference costs by utilizing dynamic model selection and query difficulty prediction with dynamic quality adjustment mechanisms, respectively. While this approach performs well within the data distribution, its generalization ability outside the distribution is suboptimal. Moreover, it requires collecting a large amount of data to create training labels for the router.

### 2.3 CoE with benchmark and Mapping

Some researchers have attempted to incorporate benchmarks datasets as training data during the CoE router training process, such as LLM-Bench Shnitzer et al. (2023). LLM-Bench reconstructs benchmark datasets to train a router model for LLM selection and demonstrates that this problem can be transformed into a series of binary classification tasks. This approach aims to predict the performance of expert models on inputs with unknown task attributes. However, LLM-Bench has low efficiency in training the router, since it requires to re-evaluate the expert model for query labels, like the single-inference CoE as shown in Figure 2 (c). Besides, the LLM-Bench router is not flexible for extension, since the router directly maps inputs to the expert model space. To alleviate this issue, MODULAR-CoE Jain et al. (2024) adopts a two-step routing strategy comprising input classification and category-to-expert mapping, which achieves improved performance while reducing computational overhead. However, MODULAR-CoE requires the additional collection of a large amount of labeled data to train the router for input classification, and to train the category-to-expert mapping. Under these circumstances, the router needs to be re-trained, if there are any changes in the pool of experts.

Distinct from LLM-Bench and MODULAR-CoE, our proposed Subject-Level Bench-CoE effectively exploits subject-level labels from benchmark evaluations, without requirement to re-evaluate the queries. In addition, the routing model of our method consists of a subject classifier and a subject-expert mapping, as shown in Figure 2 (d). Subject-expert mapping can also be directly obtained from benchmark evaluations and can be updated in real time as models evolve, without the need for retraining. The query is first classified into a subject, and then the most specialized expert is selected through this subject-expert mapping to process the query, which makes our Subject-Level Bench-CoE flexible and scalable. More importantly, our method has better generalization than LLM-Bench.

## 3 Framework of Bench-CoE

In this section, we first formalize the proposed Bench-CoE framework. Then, under this framework, we introduce the Query-Level Bench-CoE which is an abstraction of the ideas behind some existing methods Shnitzer et al. (2023); Stripelis et al. (2024); Lu et al. (2024); Ong et al. (2024).

### 3.1 Formulation of Bench-CoE

Bench-CoE consists of a set of expert models $\mathcal{M}$, a router $\mathcal{R}_\theta^N(q)$ for query $q$ routing, and a benchmark dataset $\mathcal{B}$ for training the router. The goal of Bench-CoE is to utilize the evaluation results of expert models $\mathcal{M}$ on the benchmark dataset $\mathcal{B}$ as labels for training the router $\mathcal{R}_\theta^N(q)$, thereby enabling the reasonable assignment of input query $q$ to the appropriate expert model for processing. The whole process can be formulated as:

$$o = f(\mathcal{M}, \mathcal{R}, q). \tag{1}$$

Where $o$ is the final output of Bench-CoE. $f$ represents the operation of the router $\mathcal{R}$ assigning an expert model of $\mathcal{M}$ to process the input query $q$.

$$\mathcal{M} = \{M_1, M_2, \ldots, M_N\}. \tag{2}$$

Where $N$ is the number of experts model, $M_n$ represents the $n$-th expert model which can be used to process query $q$, denoted as $M_n : q \to M_n(q)$.

The router $\mathcal{R}_\theta^N(q)$ contains learnable parameters $\theta$, and it is applied to allocate a query $q$ to an appropriate expert model for processing. It can be a NLP model or a multi-modal, etc.

$$\mathcal{R}_\theta^N(q) : q \rightarrow \{p_1, p_2, \ldots, p_N\}. \tag{3}$$

Where $p_n$ represents the probability of routing the query $q$ to expert model $M_n$ for processing.

The benchmark dataset $\mathcal{B}$ may consist of $K$ subjects and can be expressed as

$$\mathcal{B} = \{S_1, S_2, \ldots, S_K\}. \tag{4}$$

Where $S_k$ represents the $k$-th subject in the benchmark dataset $\mathcal{B}$. $|S_k|$ represents the number of queries in $S_k$, i.e. $S_k = \{q_k^1, q_k^2, \ldots, q_k^{|S_k|}\}$. For example, in the MMMU benchmark validation dataset, the Math subject consists of 30 mathematical choice questions. Based on this framework, we first formalize the Query-Level Bench-CoE.

## 3.2 Query-Level Bench-CoE

Query-Level Bench-CoE requires evaluating the performance $P$ of different expert models on each query in the benchmark dataset $\mathcal{B}$. Here, $P(M_n(q_k^i), t_k^i)$ represents the performance metric that measures the similarity between the output of the expert model $M_n$ and the ground truth label $t_k^i$ for the $i$-th query sample $q_k^i$ in the $k$-th subject $S_k$ of the benchmark dataset $\mathcal{B}$. For example, in the MMMU benchmark dataset, a Math subject choice question is evaluated by checking whether the option selected by the expert model matches the standard answer.

**Expert Models Selection**    Theoretically, to achieve the best combination results, the performance of a sufficiently large number of models should be tested on each query. However, this is highly costly in applications and makes it difficult to train an efficient router. We can directly choose these specialized expert models for combination since the expert models selected based on subject specialization from the benchmark are already the best within their respective fields. Experimental results show that this simplified approach is effective.

Once the performance of different expert models on each query in the benchmark dataset is obtained, the expert model with the best performance is used as the routing label for training the router in Query-Level Bench-CoE. The query-level routing label $t_k^i$ for query $q_k^i$ is determined as follows:

$$t_k^i = \arg\max_l P(M_l(q_k^i), t_k^i). \tag{5}$$

If multiple models answer the same query correctly, we assign the label to the model with the highest overall score in the corresponding subject.

**Query-Level Bench-CoE Inference.**    Given an input query $q$, the router $\mathcal{R}_\theta^N(q)$ of Query-Level Bench-CoE first predicts the most suitable expert model for handling the query:

$$n = \arg\max \mathcal{R}_\theta^N(q). \tag{6}$$

Where $n$ represents the index of the most suitable expert model for handling query $q$. Then, the query is fed into the selected expert model $M_n$, producing the final output of Bench-CoE:

$$o_n^q = M_n(q). \tag{7}$$

Where $o_n^q$ represents the output result of the expert model $M_n$ when processing the query $q$.

The key to this approach is pre-evaluating different expert models on each query to obtain the benchmark data labels needed for training the router. Using benchmark datasets to generate labeled data eliminates the labor-intensive process of manual annotation required for collecting additionally methods Stripelis et al. (2024); Lu et al. (2024); Ong et al. (2024). As benchmarks continue to expand, the advantages of this approach will become increasingly apparent. Query routing is more accurate within the same data distribution. However, for queries out of the training data distribution, further exploration is warranted. Furthermore, some benchmarks Wang et al. (2024); Yue et al. (2024) only provide evaluation results at the subject level rather than the query level, making it necessary to re-evaluate expert models on the benchmark datasets to obtain training data labels.

This prompts us to consider whether there exists a routing label determination method that ensures both better generalization and without computational cost. This leads to our proposed approach: Subject-Level Bench-CoE.

## 4 Subject-Level Bench-CoE

We analyze that the key to solve this challenge lies in how to reasonably determine the routing labels for queries. In benchmark tests, the performance of different expert models across different subjects can actually serve as a type of label. Subject-Level Bench-CoE is based on subject-level routing labels, and an example of Subject-Level Bench-CoE is shown in Figure 3.

The router of Subject-Level Bench-CoE consists of a subject classifier and a subject-expert mapping. The subject classifier label for the query within a subject in the benchmark dataset are assigned to that subject and some benchmarks Wang et al. (2024); Yue et al. (2024) typically include the subject information of queries automatically. The subject-expert mapping establishes a correspondence between each subject in the benchmark and the expert model that performs best in that subject. The relationship between the expert model index $n$ and the subject index $k$ is established by the mapping:

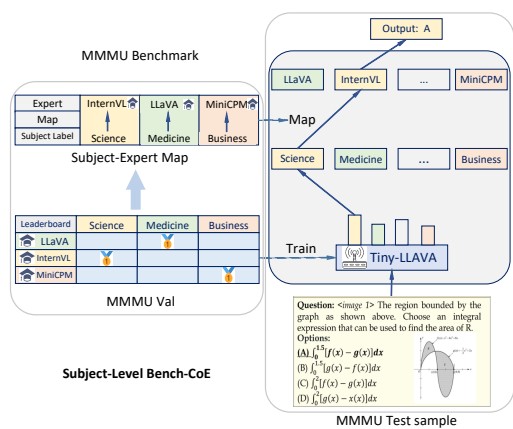

Figure 3: Subject-Level Bench-CoE first classifies the query into a corresponding subject and then leverages the subject-expert mapping derived from benchmark to facilitate the selection of the expert most proficient in handling subject task.

$$n = \arg\max_l \frac{1}{|S_k|} \sum_{i=1}^{|S_k|} P(M_l(q_k^i), t_k^i). \tag{8}$$

The retrieval of subject-expert $Map$ almost no additional effort, as the benchmark leaderboard already provides the performance results of expert models in each subject, such as Wang et al. (2024); Yue et al. (2024). The $Map$ has already been completed when the leaderboard was published, meaning we only need to select the model with the highest performance in each subject.

$$Map : \{S_1, \ldots, S_K\} \to \{M_1, \ldots, M_N\}. \tag{9}$$

$K \geq N$ indicates that a single expert model may achieve the best performance across multiple subjects. Additionally, as more expert models are developed, the models that excel in specific subjects may change over time. To avoid retraining the router, we set the number of classification neurons in the last layer of the router to be equal to the number of subject categories. Then, by adjusting the mapping between subjects and expert models, we accommodate changes in subject-specialized expert models. This simple technique enhances the flexibility of the router, allowing it to adapt dynamically to changes in expert models' subject specializations without the need for retraining.

**Subject-Level Bench-CoE Inference.** Given an input query $q$, the classifier $\mathcal{C}_\theta^K$ of Subject-Level Bench-CoE first predicts the category of the most relevant subject.

$$k = \arg\max \mathcal{C}_\theta^K(q). \tag{10}$$

Where $k$ represents the category of the most relevant subject for the input query $q$. Then, the mapping between subjects and their corresponding specialized expert models is used to determine the index of the most suitable expert model for processing the input.

$$M_n = Map(S_k). \tag{11}$$

Finally, the query is fed into the selected expert model $M_n$, producing the final output of Subject-Leve Bench-CoE:

$$o_n^q = M_n(q). \tag{12}$$

Where $o_n^q$ represents the output result of the expert model $M_n$ when processing the query $q$. From the above, it can be seen that our classifier $\mathcal{C}_\theta^N$ and mapping $Map$ together form the subject-level router $\mathcal{R}_\theta^N$.

Subject-Level Bench-CoE is more likely to correctly assign queries to the appropriate expert model compared to the Query-Level Bench-CoE. This is because, as long as the trained router can correctly classify the query into the appropriate subject category and then allocate it to the expert model proficient in that subject, it is more likely to be solved correctly compared to a non-specialized expert model. For instance, assigning a question about how to travel to Mars to a physics expert is more likely to yield a correct answer than assigning it to a literature expert.

Table 1: Naive Evaluation on MMMU.

| Model | Accuracy |
|---|---|
| MiniCPM-V-2.6 Yao et al. (2024) | 45.22% |
| InternVL2-8B Chen et al. (2024b) | 47.67% |
| LLaVA-OV-7B Li et al. (2024) | 46.67% |
| Query-Bert-Bench-CoE | 62.33% |
| **Query-TinyLLaVA-Bench-CoE** | **62.44%** |
| Subject-Bert-Bench-CoE | 51.78% |
| Subject-TinyLLaVA-Bench-CoE | 52.78% |

Table 2: In-distribution Evaluation on MMMU.

| Model | Accuracy |
|---|---|
| MiniCPM-V-2.6 Yao et al. (2024) | 39.40% |
| InternVL2-8B Chen et al. (2024b) | 44.20% |
| LLaVA-OV-7B Li et al. (2024) | 41.30% |
| Query-Bert-Bench-CoE | 41.40% |
| Query-TinyLLaVA-Bench-CoE | 41.30% |
| Subject-Bert-Bench-CoE | 44.50% |
| **Subject-TinyLLaVA-Bench-CoE** | **44.70%** |

## 5 EXPERIMENTS

We conducted extensive experiments on both multimodal and language tasks to validate the effectiveness of our proposed method. The experiments were designed to assess the performance of our Bench-CoE model against individual expert models in various settings as follows.

### 5.1 EVALUATION SCENARIOS

**Naive Evaluation.** In this scenario, we use a split of the benchmark dataset $\mathcal{B}_{val}$ to obtain labels for training the router and evaluate the performance of each expert model and the Bench-CoE framework on the same split $\mathcal{B}_{val}$.

**In-distribution Evaluation.** In this scenario, we use a split of the benchmark dataset $\mathcal{B}_{val}$ to construct labels for training the router and evaluate the performance of each expert model and the Bench-CoE framework on a different split $\mathcal{B}_{test}$.

**Out-of-distribution Evaluation.** In this scenario, we utilize a split of the benchmark dataset $\mathcal{B}_{val}^1$ to define labels for training the router and evaluate the performance of the Bench-CoE framework on a split of the other benchmark dataset $\mathcal{B}_{val}^2$. Compared to the previous two scenarios, it further improves the model's generalization ability evaluation.

**Cross-domain Evaluation.** In this scenario, we adapt our Bench-CoE framework from multimodal tasks to NLP tasks. We utilize a split of an NLP benchmark dataset $\mathcal{B}_{val}^{NLP_1}$ to define labels for training the router and evaluate the performance of the Bench-CoE framework on a split of the other NLP benchmark dataset $\mathcal{B}_{val}^{NLP_2}$. Compared to previous scenarios, this setup validates the domain generalization capability of our Bench-CoE framework.

### 5.2 NAIVE EVALUATION

To validate the effectiveness and feasibility of the Bench-CoE framework, we conducted naive evaluation on multimodal tasks. Specifically, we conducted training and testing of Bench-CoE framework on the validation split of the MMMU dataset Yue et al. (2024). We take three expert models with distinct performance strengths as representative examples (additional models can be handled in a similar manner), MiniCPM-V-2.6 Yao et al. (2024), InternVL2-8B Chen et al. (2024b), and LLaVA-OV-7B Li et al. (2024), which performed well across 30 subjects. And we evaluated and compared the performance of Bench-CoE with BERT Devlin et al. (2019) and TinyLLaVA Zhou et al. (2024) as classifiers. The results are shown in Table 1 and Figure 4a.

It can be observed that in the naive scenario, both Query-Level Bench-CoE and Subject-Level Bench-CoE achieve significant performance improvements compared to any individual expert model, with respective increases of 14.66%, 14.77%, 4.11%, and 5.11%. The performance improvement of Query-Level Bench-CoE is more pronounced.

Additionally, although Query-Level Bench CoE performs better under this setting, its training routing requires fine-grained labels, which can not be directly obtained from the leaderboard. Thus it requires an additional label processing step compared to Subject-Level Bench CoE. For the three models chosen in our experiments and the MMMU validation set containing only 900 samples, Query-Level Bench CoE takes about 20 minutes to process these labels. As the dataset size increases and the number of models grows, the time required for label processing will also increase

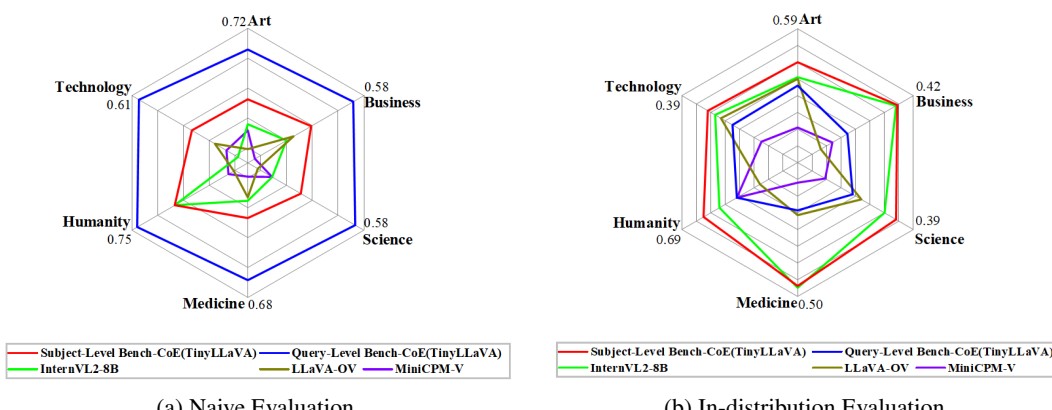

(a) Naive Evaluation.  (b) In-distribution Evaluation.

Figure 4: (a) Performance of Naive Evaluation on MMMU. (b) Performance of In-distribution Evaluation on MMMU.

significantly. In contrast, Subject-Level Bench CoE can directly obtain labels from the leaderboard without any additional time cost.

## 5.3 IN-DISTRIBUTION EVALUATION

To further validate the generalization capability of Subject-Level Bench-CoE and assess its effectiveness, we conducted in-distribution experiments on the MMMU dataset. Specifically, we conducted training and testing of Bench-CoE framework on the validation and test split of the MMMU dataset, respectively. The results are shown in Table 2 and Figure 4b.

The results of the in-distribution experiment indicate that the performance of Query-Level Bench-CoE on the MMMU dataset has deteriorated significantly, with response accuracy 2.80% and 2.90% lower than that of its respective expert models. In contrast, the proposed Subject-Level Bench-CoE maintains strong generalization in the in-distribution evaluation, achieving 0.3% and 0.5% higher response accuracy compared to its corresponding expert models. We attribute this to the enhanced generalization capability of the subject-level router, which is trained using subject-level labels, as opposed to the query-level router, which relies on query-level labels.

## 5.4 OUT-OF-DISTRIBUTION EVALUATION.

Finally, to thoroughly validate the generalization capability of the Subject-Level Bench-CoE and assess its effectiveness, we conducted out-of-distribution experiments on the MMMU and MMstar Chen et al. (2024a) dataset. Specifically, we conducted training of Bench-CoE on the validation split of the MMMU dataset and testing on the validation split of the MMstar. The results are shown in Table 3 and Figure 5a.

It can be seen that Query-Level Bench-CoE still suffers from severe generalization deficiencies in out-of-distribution scenarios, with response accuracy 3.22% and 3.6% lower than that of its respective expert models. In contrast, the proposed Subject-Level Bench-CoE demonstrates superior generalization, achieving 0.87% and 0.86% higher response accuracy compared to its corresponding expert models, even on datasets with different distributions. This further validates the Subject-Level Bench-CoE, which exhibits stronger generalization capability than the Query-Level Bench-CoE.

## 5.5 CROSS-DOMAIN EVALUATION

Furthermore, in addition to verifying the effectiveness of Subject-Level Bench-CoE for multimodal models, we also conducted cross-domain model validation. Specifically, we evaluated the effectiveness of the Bench-CoE framework on Large Language Models (LLMs) through Out-of-Distribution (OOD) experiments on selected NLP datasets such as MMLU-Pro Wang et al. (2024) and Big-Bench-Hard Suzgun et al. (2023). Analogous to the multimodal processing approach, we select four expert models with complementary strengths across different domains as illustrative examples: Qwen2-7B-InstructYang et al. (2024), Gemma-2-9b-itTeam (2024), Mathstral-7B-v0.1Jiang et al. (2023a), and Llama-3-Smaug-8BMeta (2024). The results are shown in Table 4 and Figure 5b.

Table 3: Performance of the Out-of-distribution Evaluation on MMMU and MMStar val.

| Model | Accuracy |
|---|---|
| MiniCPM-V-2.6 Yao et al. (2024) | 54.33% |
| InternVL2-8B Chen et al. (2024b) | 59.22% |
| LLaVA-OV-7B Li et al. (2024) | 55.86% |
| Query-Bert-Bench-CoE | 56.00% |
| Query-TinyLLaVA-Bench-CoE | 55.62% |
| **Subject-Bert-Bench-CoE** | **60.09%** |
| Subject-TinyLLaVA-Bench-CoE | 60.08% |

Table 4: Performance of the Cross-domain Evaluation on MMLU-Pro and BBH.

| Model | Accuracy |
|---|---|
| Qwen2-7B-Instruct Yang et al. (2024) | 59.44% |
| Gemma-2-9b-it Team (2024) | 65.10% |
| Mathstral-7B-v0.1 Jiang et al. (2023a) | 66.35% |
| Llama-3-Smaug-8B Meta (2024) | 63.62% |
| Query-Bert-Bench-CoE | 67.07% |
| **Subject-Bert-Bench-CoE** | **69.91%** |

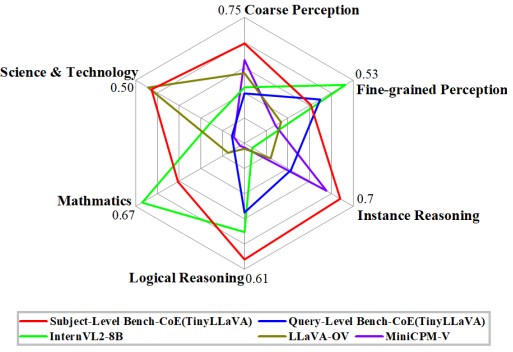

(a) Out-of-distribution Evaluation

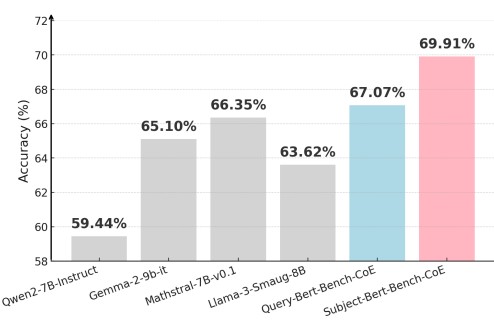

(b) Cross-domain Evaluation.

Figure 5: (a) Performance of Out-of-distribution Evaluation on MMstar. (b) Performance of Cross-domain Evaluation on MMLU-Pro val and Big-Bench-Hard).

The results of the cross-domain experiment indicate that the Bench-CoE framework remains effective for LLMs in out-of-distribution scenarios, with response accuracy still 0.72% and 3.56% higher than that of the corresponding expert models. Moreover, the improvement of the Subject-Level Bench-CoE is particularly significant. This also validates the generalization of our framework.

## 6 LIMITATIONS AND FUTURE WORK

Although our proposed Bench-CoE demonstrates favorable performance in terms of generalization, flexibility, and avoiding the need for data annotation, it still has the following limitation: Diversification of Model Capabilities. Previous experiments have demonstrated that CoE models achieve optimal effectiveness only when each expert model exhibits distinct specialized capabilities. The presence of a minority of models that hold significant leads across all capabilities will diminish the effectiveness enhancement of such collaborative systems. In future research, we plan to explore more balanced expert selection strategies and adaptive mechanisms to mitigate the impact of dominant experts, thereby enhancing the robustness and scalability of Bench-CoE in broader multi-task scenarios.

## 7 CONCLUSION

We propose the Bench-CoE framework, designed to facilitate effective collaboration between experts in LMMs and LLMs by leveraging benchmark evaluation results. The framework consists of the Query-Level Bench-CoE, abstracted from existing methodologies, and the Subject-Level Bench-CoE, which we introduce to enhance both the performance and scalability of the framework. Extensive experiments on multimodal and language tasks have validated the feasibility of our framework and demonstrated the generalization ability and efficiency of the Subject-Level Bench-CoE. We hope that Bench-CoE can offer a paradigm for integrating diverse experts, enhancing adaptability while reducing reliance on labeled data. It is expected to inspire the development of more robust and scalable expert collaboration systems and their application in real-world multi-task scenarios.

## ETHICS STATEMENT

This research strictly adheres to the ICLR Code of Ethics. Our work does not involve human subjects, nor does it use datasets containing sensitive, private, or discriminatory content. All datasets employed are publicly available and used in compliance with their licenses and release practices. The methods and conclusions of this research do not contain potential malicious applications, and we have carefully evaluated and avoided possible negative societal impacts. There are no conflicts of interest or inappropriate sponsorship involved, and all experiments and results comply with research integrity and academic standards.

## REPRODUCIBILITY STATEMENT

We have made every effort to ensure the reproducibility of our results. The paper and appendix include detailed descriptions of the model architecture, algorithmic procedures, and experimental settings. All datasets used are publicly available, and the preprocessing steps are documented in the supplementary materials. We have released our complete source code and experimental configurations, allowing other researchers to independently reproduce our experiments and main findings.

## LLM USAGE STATEMENT

A large language model (LLM) was employed solely for grammatical error checking during the preparation of this manuscript. The LLM was not used for generating research ideas, designing experiments, analyzing results, or writing substantive scientific content. All methodological and experimental contributions are the authors' own work.

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

# A MODELS AND DATASETS

## A.1 MULTIMODAL MODELS

In the evaluation of Bench-CoE on multimodal tasks, we employ TinyLLaVA and BERT as the subject classifiers for multimodal and textual inputs, respectively. Furthermore, we used MiniCPM-V-2.6, InternVL2-8B, and LLaVA-OV-7B as expert models.

TinyLLaVA is a multimodal model series that takes text and image inputs and outputs textual responses. In our experiments, we use TinyLLaVA-Phi-2-SigLIP-3.1B, which employs Phi-2 as the language model and SigLIP as the image encoder. Based on this model, we add a linear layer to enable classification.

MiniCPM-V-2.6 is a multimodal language model developed to integrate visual processing with natural language understanding. With 2.6 billion parameters, this model is a compact version of the larger CPM series, designed to efficiently handle tasks that require the synthesis of textual and visual data. MiniCPM-V-2.6 excels in image captioning, visual question answering, and other applications where joint understanding of text and image is critical. Its training regimen includes diverse datasets from both textual and visual domains, ensuring robust performance across a variety of multimodal challenges.

InternVL2-8B is an 8 billion parameter model specifically designed for video-language tasks. Developed to bridge the gap between dynamic visual content and language, InternVL2-8B can analyze and generate descriptions for video data, making it highly suitable for applications such as automated video captioning, video content analysis, and interactive video-based learning systems. Its architecture allows for deep understanding of temporal video sequences in conjunction with textual descriptions, providing state-of-the-art results in video understanding tasks.

LLaVA-OV-7B, standing for Language and Vision Analysis - OmniVision, is a 7 billion parameter language model that specializes in comprehensive visual and textual interpretation. This model integrates advanced vision capabilities with natural language processing to perform tasks like detailed image analysis, multimodal translation, and cross-modal information retrieval. LLaVA-OV-7B is trained on a vast array of multimodal data sources, enabling it to effectively understand and generate content that requires the amalgamation of visual cues with textual data.

## A.2 MULTIMODAL TASKS

MMMU is a comprehensive dataset designed for evaluating the performance of multimodal models across tasks that require simultaneous understanding of text, image, and sometimes audio content. This dataset includes challenges such as cross-modal retrieval, multimodal reasoning, and synchronizing visual content with textual descriptions. MMMU aims to simulate real-world scenarios where multiple types of data must be integrated and interpreted together.

MMStar is a multimodal dataset focused on the interplay between visual and textual data in entertainment and media contexts. It includes annotated images and videos from various media sources, coupled with descriptive texts and contextual information. The dataset is utilized for tasks such as multimedia content summarization, sentiment analysis, and thematic classification, testing a model's ability to navigate and interpret complex media-rich environments.

## A.3 LANGUAGE MODELS

In the cross-domain evaluation of Bench-CoE on NLP tasks, we employ BERT as the subject classifier for textual inputs. Additionally, we utilize Qwen2-7B-Instruct, Gemma-2-9B-IT, Mathstral-7B-v0.1, and Llama-3-Smaug-8B as expert models.

BERT (Bidirectional Encoder Representations from Transformers) is a pre-trained deep learning model based on the Transformer architecture, designed to capture bidirectional contextual representations of text. By leveraging a masked language modeling (MLM) and next sentence prediction (NSP) pre-training strategy, BERT enables robust language understanding across various natural language processing (NLP) tasks. It has demonstrated state-of-the-art performance in tasks such

as text classification, named entity recognition, and question answering, making it a fundamental model in modern NLP research and applications.

Qwen2-7B-Instruct is an instruction-focused language model developed by Qwen Technology. Designed to excel in various natural language understanding tasks, this model utilizes an optimized decoding strategy to enhance performance. With 7 billion parameters, it is well-suited for complex text comprehension and generation tasks, especially in Chinese contexts. Qwen2-7B-Instruct is particularly effective for instruction-responsive tasks such as content creation, information extraction, and dialogue systems.

Gemma-2-9b-it is a large language model developed by Gemma Technologies with 9 billion parameters, tailored for the information technology (IT) sector. Its training data encompasses a vast array of technical documents, programming guides, and texts from open-source projects. This model excels in understanding and generating highly specialized IT content, making it ideal for applications in technical support, documentation automation, and code parsing.

Mathstral-7B-v0.1 is a language model focused on solving mathematical problems, developed by the Mathstral team. With 7 billion parameters, its training includes extensive mathematical educational materials and real-world problem-solving examples. Mathstral-7B-v0.1 is designed to aid in mathematical education, automated problem-solving, and mathematical research, particularly effective for complex mathematical questions and theoretical discussions.

Llama-3-Smaug-8B is the latest large language model from the Llama team, featuring 8 billion parameters. It has been extensively pre-trained across multiple languages and domains to provide broad knowledge coverage and deep semantic understanding. Llama-3-Smaug-8B emphasizes performance in complex linguistic reasoning, long-form text generation, and multi-domain knowledge integration, suitable for advanced natural language processing tasks such as text summarization, language translation, and cross-domain knowledge-based question answering.

## A.4 LANGUAGE TASKS

MMLU-Pro is an extension of the original MMLU dataset, designed to evaluate language models on professional-level topics across a wide array of subjects. This dataset includes complex questions that require not only language understanding but also domain-specific knowledge, ranging from medicine and law to engineering and the arts. MMLU-Pro aims to test the depth and breadth of a model's understanding of advanced topics, making it a rigorous benchmark for language comprehension.

Big-Bench-Hard is a subset of the broader BIG-bench dataset specifically curated to challenge the capabilities of language models with particularly difficult tasks. This dataset includes a variety of language-based tasks such as analogical reasoning, complex problem-solving, and advanced comprehension challenges that go beyond the typical capabilities of standard language models, pushing the limits of what AI can understand and process in textual form.

## B EXPERIMENT DETAILS

### B.1 MULTIMODAL EXPERIMENT

MMMU and MMStar are currently among the most comprehensive multimodal benchmarks, encompassing tasks such as cross-modal retrieval and multimodal reasoning. To thoroughly evaluate the performance of Bench-CoE on multimodal tasks, we designed experiments in three phases: naive test, in-distribution test, and out-of-distribution test.

In the naive test phase, we used the MMMU dataset for both training and testing the Bench-CoE router. The subset of MMMU was utilized for both training and evaluation. This phase primarily aimed to verify the basic feasibility of Bench-CoE in task allocation for multimodal tasks. By leveraging query-level and subject-level routing strategies, Bench-CoE significantly outperformed individual models, demonstrating its effectiveness in task allocation. The query-level router provided finer-grained task assignments, while the subject-level router exhibited stronger overall robustness.

In the in-distribution test phase, the test set of the MMMU dataset was used for training, and the validation set was used for evaluation. This setup ensured a clear separation between training and testing data while maintaining consistency in data distribution. The Bench-CoE router effectively allocated tasks to the most suitable expert models based on the input, showcasing its strong adaptability for tasks within the same distribution.

In the out-of-distribution test phase, the Bench-CoE router was trained on the validation set of the MMMU dataset and tested on the MMStar dataset. The MMStar is a multimodal dataset focus on the interplay between visual and textual data in entertainment and media contexts, presenting challenges to the model's generalization capabilities. The experiments demonstrated that the subject-level router remained effective in handling tasks with significant distributional differences, validating the adaptability and robustness of Bench-CoE. In contrast, the query-level router showed slightly reduced performance on new data distributions, likely due to overfitting.

These experimental results indicate that Bench-CoE effectively integrates the strengths of different models, achieving outstanding performance in both in-distribution and out-of-distribution tasks. This approach provides a solid foundation for further research on collaborative mechanisms in multimodal models.

## B.2 LANGUAGE EXPERIMENT

Due to the current limitations in large model evaluation techniques, there is a relative scarcity of benchmarks and datasets specifically tailored to academic disciplines. To the best of our knowledge, only the MMLU-Pro and Big-Bench-Hard datasets include manually annotated discipline-specific labels. This poses significant challenges to the experimental design of our Bench-CoE model. To thoroughly evaluate the generality of Bench-CoE, we conducted this test:

In the out-of-distribution test phase, we selected datasets with strongly defined discipline-specific features: the MMLU-Pro dataset as the training set and the Big-Bench-Hard dataset as the test set. Specifically, we trained the Bench-CoE router on the MMLU-Pro dataset and evaluated it on the Big-Bench-Hard dataset. By testing across different datasets with distinct data distributions, and with both training and test sets exhibiting clear discipline-specific characteristics, this phase allowed us to thoroughly validate the cross-domain generalization capability of the Bench-CoE model at both the query-level and subject-level.

## B.3 COMPARATIVE EVALUATION OF BENCH-COE AND LARGE-SCALE LMMS

Furthermore, to validate the performance of our Bench-CoE framework in comparison with larger independent models, we selected models with over 10 billion parameters from the leaderboard, which significantly exceed the 7 billion parameter expert models chosen by our Bench-CoE framework at each selection step. These models were tested on the validation split of the MMMU dataset and the result is shown in Tab. 5 and Fig. 6.

Table 5: Performance of Bench-CoE and Large-Scale LMMs on the validation split of MMMU.

| Model | Accuracy |
|---|---|
| Math-LLaVA-13B | 38.3% |
| Yi-VL-34B | 45.9% |
| Subject-Level Bench-CoE (BERT) | 50.78% |
| **Subject-Level Bench-CoE (TinyLLaVA)** | **52.00%** |

Through observation, it can be seen that the proposed Bench-CoE framework enables the collaboration of multiple small-parameter models to surpass large-parameter models in terms of performance.

## C SCALABILITY OF BENCH COE

In Bench-CoE, particularly in the subject-level Bench-CoE, we leverage the best-performing LLM for each domain as the routing target. By directing as many questions as possible within a given

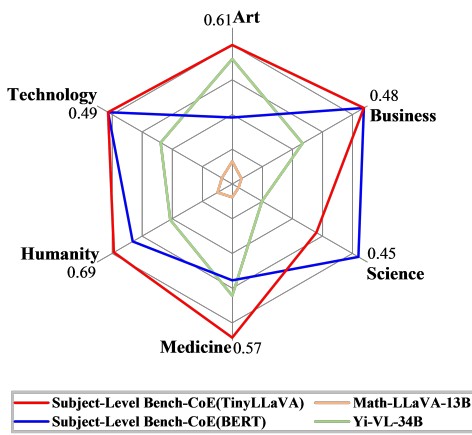

Figure 6: Performance of each expert model and CoE across subjects on MMstar in the Out-of-distribution Evaluation.

subject to the "best" expert for inference, we enhance the overall accuracy of the model. However, with the rapid evolution of large language models, accompanied by the introduction of new datasets, novel models, and updated evaluation methods, the leaderboard rankings of LLMs change frequently. Under such circumstances, a fixed routing strategy in the CoE model cannot accommodate newly emerging models or adapt to shifting data distributions.

To address this limitation and improve the scalability of Bench-CoE, we designed a subject-expert mapping mechanism. Instead of directly routing inputs to a fixed best-performing model in a domain, we first classify the given input into a specific subject type. Then we leverage the subject-expert mapping to route the input to the most suitable model for that domain. This approach significantly enhances the scalability of Bench-CoE, allowing it to flexibly adapt to rapidly evolving expert models advancements by dynamically adjusting the mapping and updating routing rules.

# D    SCENARIOS UNSUITABLE FOR COE

**Lack of Diversification in Model Capabilities**   In our experiments with the Bench-CoE model, we selected a wide range of LLMs as candidate models and conducted extensive testing. Through these tests, we identified a common challenge in the CoE field: the issue of expert capability diversity. Specifically, this problem arises when a candidate expert lacks capability diversity on the given dataset — either significantly outperforming or underperforming all other candidate expert. Such cases negatively impact the overall performance of the CoE models, as the router is forced to route all queries either exclusively to or completely away from this model to achieve optimal results. This creates a significant challenge for training the router.

Looking ahead, we believe this issue can be mitigated with the development of dynamic routing strategies and adaptive candidate LLM selection mechanisms. These advancements will enable the CoE model to better handle capability imbalances among candidate LLMs, paving the way for more robust and flexible routing solutions.