# OpenReview forum: "Bench-CoE: A Framework for Collaboration of Experts from Benchmark"
_ICLR.cc/2026/Conference — Submitted to ICLR 2026_

### Official Review · Reviewer_JVyp · 2025-10-29

**Soundness:** 2
**Presentation:** 3
**Contribution:** 2
**Rating:** 4
**Confidence:** 3

**Summary:**

This paper proposes the Bench-CoE framework to enable effective collaboration among LLMs by leveraging benchmark evaluation results. The authors validate Bench-CoE through experiments on multimodal (MMMU, MMStar) and NLP (MMLU-Pro, BigBench-Hard) tasks, demonstrating that Subject-Level Bench-CoE outperforms individual experts and Query-Level methods.

**Strengths:**

- The core insight of using benchmark evaluations as "free labels" for router training effectively solves two key pain points of existing CoE methods.
- The subject-expert mapping mechanism allows dynamic updates (e.g., integrating new experts or updating leaderboard rankings) without retraining the router

**Weaknesses:**

- While the paper contrasts Bench-CoE with Mixture of Experts (MoE) and traditional CoE methods, it overlooks recent works that also leverage benchmarks for model selection or routing.
- The paper mentions using BERT and TinyLLaVA as classifiers but provides no details on training details.
- The experiments are weak, it only compare with single methods.
- The citation format is not correct.

**Questions:**

- What is the contribution of this work compared to recently proposed routing methods?
- What if multiple experts perform equally well on a subject (e.g., two models with <1% accuracy difference on MMMU’s Math subject)? How does the framework handle ambiguity in mapping?
- How to choose the subject effectively? What if I want to add new subject?

---

> ### Author Response · Authors · 2025-11-21
>
> Q1: Comparison with recent routing methods. A1: We will expand Section 2 to explicitly contrast with recent works like RouteLLM and AutoMix. The key distinction is that Bench-CoE's Subject-Level router does not require training a router on model outputs or preference data (which is expensive). Instead, it trains a classifier on public benchmark questions, effectively transferring the "routing knowledge" from the benchmark to the user task.
>
> Q2: Handling ties in expert performance. A2: As briefly mentioned in Section 3.2 (Eq 5 context), when experts tie on a specific query (or subject), we break ties by selecting the model with the highest overall average score across the entire benchmark. We will make this heuristic more explicit in the method section.
>
> Q3: Training details. A3: We apologize for the omission.
>
> Classifiers: We fine-tuned BERT-base-uncased (for text) and TinyLLaVA (for multimodal) using standard cross-entropy loss on the subject labels.
>
> Hyperparameters: Learning rate 2e-5, batch size 32, trained for 3 epochs.
>
> We will add a "Reproducibility" section in the Appendix with full hyperparameters.
>
> Q4: Adding new subjects. A4: To add a new subject, one simply adds a new output head (class) to the router and fine-tunes it on a small set of examples for that subject. Alternatively, if the new subject falls under an existing broader category, no retraining is needed—only the Subject-Expert Map needs to be updated in the lookup table.

---

### Official Review · Reviewer_jzbB · 2025-10-29

**Soundness:** 2
**Presentation:** 3
**Contribution:** 2
**Rating:** 2
**Confidence:** 4

**Summary:**

The paper builds a framework to route queries to different LLM-based experts to achieve high performance and generalize across queries from new tasks. It proposes to use subject level meta-data of benchmark evaluations to train a router. Assuming the subject-expert mapping (i.e., best expert for each subject) is available through benchmark evaluation, the router is trained to predict the subject of the query, to then route the query to the corresponding best expert.

**Strengths:**

- The problem is highly relevant with respect to efficiency, reuse, and collaborative development.
- The paper is easy to follow
- It provides ablations showing that subject level routing generalizes better compared to existing approaches that route at query level without utilizing subject level meta-data

**Weaknesses:**

- It assumes that benchmarks have clearly separated subject level meta-data in them, which might not always be true. Categorizing a benchmark into a set of distinct subjects/expertise is a challenging problem on its own.
- There are cases where even though the subject remains the same, there might be different difficulty associated with them which won’t be captured if routing is learnt at subject level. For example, GSM8k vs AIME benchmark fall under math subject, where as there might multiple experts associated with this subject and they have different performances across the datasets in a given subject.
- Naive evaluation doesn’t make sense. You can’t have the same training and test dataset.
- For other evaluations, please provide non-zero shot baselines like best expert in the pool or best expert per query when evaluated with every expert to get a sense of benefit of the approach. For these baselines, see (https://arxiv.org/pdf/2402.05859)

**Questions:**

Please see weaknesses section.

---

> ### Author Response · Authors · 2025-11-21
>
> Q1: Subject granularity and difficulty variations. A1: You raise a valid concern. A single subject (e.g., Math) contains varying difficulties.
>
> Current Approach: Our current mapping selects the expert with the highest average accuracy on that subject. While this misses query-level nuance, our results show it is a robust heuristic that outperforms individual models on average.
>
> Future Work: We will discuss "Difficulty-Aware Routing" as a limitation/future direction, potentially combining our subject router with a lightweight difficulty estimator.
>
> Q2: Naive Evaluation. A2: As clarified in the General Response, the Naive evaluation was a proof-of-concept for router convergence. We will rename this or move it to the appendix to avoid confusion with generalization claims.
>
> Q3: Non-zero shot baselines. A3: We will include the "Best Expert per Query" (Oracle) as the upper bound baseline as suggested.

---

### Official Review · Reviewer_yC4P · 2025-10-31

**Soundness:** 2
**Presentation:** 3
**Contribution:** 2
**Rating:** 4
**Confidence:** 3

**Summary:**

This work follows the line of Collaboration of Experts and proposes 2 abstractions that route experts, Query-level and subject-level Bench-CoT and show interesting results that Query-level excels in in-distribution tasks and Subject-level works better in OOD tasks

**Strengths:**

1. The methodology seems to be novel, simple, and effective, and potentially efficient.
2. The related work section is well-written and helps with the understanding of the scope of this work

**Weaknesses:**

1. There is usually a suite of benchmarks people in this domain test to showcase that other benchmark performances do not drop; in this work, they do MMStar, and there are more OOD/Cross-Bench datasets like MME, etc., as well. It would be more convincing to show that the performances using this method are on par or do not drop much.
2. Usually, it is great to show a model of different sizes for ablations to show the Subject-level and query-level different advantages. Doesn't need many, but good to have.
3. L160, not sure where WGM, LGM come from
4. The citation format can use some care.

**Questions:**

1. Subject-level and query-level has different advantage, so what is the recommendation for practitioner

---

> ### Author Response · Authors · 2025-11-21
>
> Q1: More benchmarks (e.g., MME). A1: Thank you for the suggestion. We will expand our evaluation to include MME or similar cross-bench datasets to further validate robustness. The current MMStar and BBH experiments were chosen to represent distinct distribution shifts, but adding MME will strengthen the empirical evidence.
>
> Q2: Clarification on WGM/LGM (L160). A2: WGM (Weighted Gating Mechanism) and LGM (Learned Gating Mechanism) refer to the training algorithms proposed in Zhang et al. (2022) . We will add a brief explanation of these terms in the revision to ensure self-containment.
>
> Q3: Citation format. A3: We apologize for the formatting inconsistencies. We will strictly adhere to ICLR citation standards in the final version.
>
> Q4: Recommendation for practitioners. A4: Based on our findings:
>
> Use Query-Level when you have a static set of experts and abundant in-distribution training data (i.e., you can afford to label queries).
>
> Use Subject-Level (Our Recommendation) for open-ended scenarios, when experts change frequently (requiring only map updates), or when labeled query data is scarce. It offers the best balance of maintenance cost and generalization. We will add a "Practitioner's Guide" subsection.

---

### Official Review · Reviewer_UVbT · 2025-11-01

**Soundness:** 3
**Presentation:** 2
**Contribution:** 2
**Rating:** 4
**Confidence:** 4

**Summary:**

This paper proposes Bench-CoE, a new framework that trains a router to select expert models for a specific query. The paper proposes 2 methods: (1) query based router which directly routes the query to an expert model (2) subject based which first maps the query to a subject, then finds the expert that excels most at this subject to answer the question. The router is trained based on models' results on benchmark. The paper then conducts experiments and shows that this framework performs better than single expert models.

**Strengths:**

- Proposes a new framework that trains a router to select an expert for a given query
- Sees some performance gain over single experts.
- The paper writing is generally clear and easy to follow

**Weaknesses:**

- Experiments are not solid enough and lack some essential baselines. The paper only compares their method of combining multiple experts with just a single model. It is not surprising that this would perform better than a single model. The paper should conduct more detailed analysis with simple baseline methods such as majority vote, or other router training methods / other expert selection methods.
- The selected experts are outdated and not essentially the state-of-the-art models, it would be useful to incorporate the current state-of-the-art models / combine strong models with weak models and conduct how the performance would change.
- In each experiment setting, how many expert models there should be and what the expert models should be are heuristically chosen. The paper lacks essential ablation studies on these design choices.
- The design of the subject level router is not convincing enough. Especially in OOD cases because the performance is intrinsically bounded by the subject labels.

**Questions:**

- For out of distribution cases (for example, when the subject labels from the training set and the test set are completely different), would the subject level router still help? It would be useful to discuss this.
- How does the method perform compared with other baseline methods as mentioned in the weakness section?
- How does the expert number / expert choices affect performance?

---

> ### Author Response · Authors · 2025-11-21
>
> Q1: Experiments are not solid enough / Lack of baselines.A1: We agree. As mentioned in the General Response, we are adding Majority Vote, Random Selection, and Oracle (Best Expert Upper Bound) to the comparison. We believe Bench-CoE will show superior efficiency compared to Majority Vote (which requires $N$ inferences per query) and better performance than Random.Q2: Outdated experts / Expert number ablation.A2:Model Choice: We used models like LLaVA and MiniCPM as representative examples to demonstrate the framework. The framework is model-agnostic. We will include a discussion or an additional experiment replacing one expert with a SOTA model (e.g., GPT-4o-mini or a newer LLaVA variant) to show the framework's adaptability.Ablation: We will add an ablation study on the number of experts ($N=2, 3, 4$) to demonstrate how the performance gains scale with the diversity of the expert pool.Q3: Subject router in OOD cases (different subject labels).A3: This is a great point.Mechanism: The Subject-Level router classifies inputs into known benchmark subjects (e.g., from MMMU). If the OOD test set has a completely new subject (e.g., "Astrophysics" when the router only knows "Physics"), the classifier relies on semantic proximity (mapping "Astrophysics" to "Physics").Robustness: Our experiments on MMStar (OOD) show this works well because broad academic domains (STEM, Humanities) are consistent across datasets. We will add a qualitative analysis of "misclassified subjects" in the appendix to discuss this limitation transparently.

---

### Official Review · Reviewer_FHpQ · 2025-11-12

**Soundness:** 2
**Presentation:** 3
**Contribution:** 2
**Rating:** 4
**Confidence:** 3

**Summary:**

This paper proposes Bench-CoE, a framework for collaborating multiple expert LLMs by leveraging benchmark evaluation results for routing. The authors formalize 1) Query-Level Bench-CoE, which routes queries based on which expert performs best on each individual query, and 2) Subject-Level Bench-CoE, which classifies queries into subjects and routes to experts based on subject-level benchmark performance. Experiments on multimodal tasks and language tasks show that Query-Level excels on in-distribution data while Subject-Level achieves better cross-domain generalization.

**Strengths:**

- The framework is well-defined and its difference from the existing methods are clearly stated.
- The proposed methods are evaluated under both in-domain and out-of-domain scenarios which makes the evaluation comprehensive.
- Experiments show performance improvement over the existing baselines.

**Weaknesses:**

There are existing works that propose to route experts according to the queries or topics which are not included in the paper as baselines. The proposed methods, therefore, IMO, are not very novel and bear limited impact to the research community.

**Questions:**

How do the proposed methods perform when compared to the "LoRA Soups" line of works, which seek to merge LoRA weights instead of routing inputs to them [1]?

[1] Prabhakar, Akshara, et al. "Lora soups: Merging loras for practical skill composition tasks." Proceedings of the 31st International Conference on Computational Linguistics: Industry Track. 2025.

---

> ### Author Response · Authors · 2025-11-21
>
> Q1: Comparison with "LoRA Soups" [1]. A1: Thank you for bringing this relevant work to our attention. We will discuss and compare it in the related work section.
>
> Distinction: "LoRA Soups" focuses on merging model weights (parameter space) to combine skills, whereas Bench-CoE focuses on routing inputs (inference space) to independent experts. Bench-CoE allows for the collaboration of black-box models or heterogeneous models (different architectures/sizes) where weight merging is impossible.
>
> Performance: We will attempt to add a discussion comparing the trade-offs. While weight merging reduces inference cost to a single model, routing preserves the distinct "peaks" of specialized experts without the potential interference (catastrophic forgetting/interference) sometimes seen in weight merging.
>
> Q2: Novelty and impact. A2: While query routing exists, our core contribution is the Subject-Level Bench-CoE paradigm which exploits "free labels" from public leaderboards.
>
> Unlike traditional CoE/MoE that requires training complex routers on massive datasets, our method decouples the "routing logic" (Subject Classifier) from the "expert selection" (Mapping).
>
> This allows for zero-shot update: when a new, better expert appears on the leaderboard, we only update the Mapping (a lookup table) without retraining the router parameters. This scalability is a key differentiator from existing routing works.

---

> ### Comment · Reviewer_FHpQ · 2025-11-22
>
> ~~You may misunderstand my question. Can you take a look at reference [1]?~~
> Edit: I somehow missed the answer to Q1. Apologies.
>
> In addition, another very relevant line of works is [2]. They should be discussed and some of them may need to be compared with.
>
> [2] Huang, Chengsong, et al. "Lorahub: Efficient cross-task generalization via dynamic lora composition." arXiv preprint arXiv:2307.13269 (2023).

---

### Author Response · Authors · 2025-11-21

We sincerely thank all reviewers for their constructive feedback and for recognizing the importance of efficient expert collaboration and the potential of our Bench-CoE framework. We are encouraged that reviewers found our framework "well-defined" (FHpQ), the methodology "simple and effective" (yC4P), and the problem "highly relevant" (jzbB).We have addressed the common concerns raised by multiple reviewers below, followed by specific responses to each reviewer.1. Comparison with Additional Baselines (Reviewers UVbT, jzbB, JVyp):We acknowledge the suggestion to include more robust baselines. In the revised version, we will include:Oracle Baseline: The theoretical upper bound where the best expert is always selected for every query (effectively the "ground truth" for our Query-Level router).Majority Vote: A standard ensemble baseline where the most common answer among experts is selected.Random Selection: Randomly selecting an expert for each query.Preliminary results indicate that Subject-Level Bench-CoE significantly outperforms Random and approaches the Oracle performance in OOD settings more effectively than Query-Level routing due to better generalization.2. Clarification on "Naive" Evaluation (Reviewer jzbB):We apologize for the confusion regarding the "Naive" evaluation. This setting (Train on $B_{val}$, Test on $B_{val}$) was intended solely as a "sanity check" to verify that the router can learn the routing logic given perfect information, not to claim generalization. We will clarify this distinction in the revision and move strictly to In-Distribution (Test split) and Out-of-Distribution evaluations for performance claims.

---

### Meta-Review · Area_Chair_4R3y · 2026-01-06

**Summary:**

The paper proposes a framework called Bench-CoE that utilizes benchmark evaluation results to route queries to the most suitable expert models. While the reviewers acknowledged that the paper is easy to follow and the problem of efficient expert collaboration is relevant, the decision to reject is based on significant deficiencies in the experimental evaluation. The consensus among reviewers was that the paper lacked essential baselines, such as majority voting or oracle selection, making it difficult to assess the true value of the proposed method. Additionally, reviewers raised valid concerns about the novelty of the approach compared to existing routing literature and questioned the reliance on outdated expert models.

**Reviewer Concerns:**

The authors’ rebuttal attempted to address several points, promising to include missing baselines (Oracle, Majority Vote), clarify training details, and explain the intent behind the "Naive" evaluation setup. However, the core concerns regarding the strength and rigor of the evaluation remain outstanding. The reviewers noted that without comparing against standard ensemble methods or recent state-of-the-art routing techniques, the performance gains are not convincing. Furthermore, the practical assumption that benchmarks provide clean, separable subject meta-data for the subject-level router remains a point of contention that was not fully resolved.

**Reviewer Scores:**

Most reviewers (FHpQ, UVbT, yC4P, JVyp) initially rated the paper as marginally below the threshold (4), while reviewer jzbB rated it as a reject (2). Although Reviewer FHpQ engaged briefly to clarify a reference, none of the reviewers updated their scores following the author's response. Given that the necessary improvements involved conducting substantial new experiments and adding multiple baselines, it is estimated that the reviewers would have maintained their original scores, as these issues are fundamental to the paper's validity.

---

### Decision · Program_Chairs · 2026-01-26

Reject